# Efficient Biocatalytic Synthesis of Dihalogenated Purine Nucleoside Analogues Applying Thermodynamic Calculations

**DOI:** 10.3390/molecules25040934

**Published:** 2020-02-19

**Authors:** Heba Yehia, Sarah Westarp, Viola Röhrs, Felix Kaspar, Robert T. Giessmann, Hendrik F.T. Klare, Katharina Paulick, Peter Neubauer, Jens Kurreck, Anke Wagner

**Affiliations:** 1Chair of Bioprocess Engineering, Faculty III Process Sciences, Institute of Biotechnology, Technische Universität Berlin, Straße des 17. Juni 135, 10623 Berlin, Germany; HEBAYEHYA@hotmail.com (H.Y.); sarah.westarp@bionukleo.com (S.W.); f.kaspar@tu-braunschweig.de (F.K.); rgiessmann@gmail.com (R.T.G.); katharina.paulick@tu-berlin.de (K.P.); peter.neubauer@tu-berlin.de (P.N.); 2Chemistry of Natural and Microbial Products Department, Pharmaceutical and Drug Industries Research Division, National Research Centre, Dokki, 12622 Cairo, Egypt; 3BioNukleo GmbH, Ackerstr. 76, 13355 Berlin, Germany; 4Chair of Applied Biochemistry, Faculty III Process Sciences, Institute of Biotechnology, Technische Universität Berlin, Straße des 17. Juni 135, 10623 Berlin, Germany; viola.roehrs@tu-berlin.de (V.R.); jens.kurreck@tu-berlin.de (J.K.); 5Faculty II Mathematics and Natural Sciences, Institute of Chemistry, Technische Universität Berlin, Strasse des 17. Juni 135, 10623 Berlin, Germany; hendrik.klare@tu-berlin.de

**Keywords:** cytostatics, dihalogenated nucleoside analogue, yield prediction, thermostable nucleoside phosphorylase, thermodynamic calculations, leukemic cell line

## Abstract

The enzymatic synthesis of nucleoside analogues has been shown to be a sustainable and efficient alternative to chemical synthesis routes. In this study, dihalogenated nucleoside analogues were produced by thermostable nucleoside phosphorylases in transglycosylation reactions using uridine or thymidine as sugar donors. Prior to the enzymatic process, ideal maximum product yields were calculated after the determination of equilibrium constants through monitoring the equilibrium conversion in analytical-scale reactions. Equilibrium constants for dihalogenated nucleosides were comparable to known purine nucleosides, ranging between 0.071 and 0.081. To achieve 90% product yield in the enzymatic process, an approximately five-fold excess of sugar donor was needed. Nucleoside analogues were purified by semi-preparative HPLC, and yields of purified product were approximately 50% for all target compounds. To evaluate the impact of halogen atoms in positions 2 and 6 on the antiproliferative activity in leukemic cell lines, the cytotoxic potential of dihalogenated nucleoside analogues was studied in the leukemic cell line HL-60. Interestingly, the inhibition of HL-60 cells with dihalogenated nucleoside analogues was substantially lower than with monohalogenated cladribine, which is known to show high antiproliferative activity. Taken together, we demonstrate that thermodynamic calculations and small-scale experiments can be used to produce nucleoside analogues with high yields and purity on larger scales. The procedure can be used for the generation of new libraries of nucleoside analogues for screening experiments or to replace the chemical synthesis routes of marketed nucleoside drugs by enzymatic processes.

## 1. Introduction

The standard production route for nucleoside analogues in the pharmaceutical industry is still chemical synthesis. It usually involves protection–deprotection steps and the use of hazardous chemicals and solvents. Furthermore, chemical synthesis routes for purine nucleoside analogues usually show low selectivity and finally lead to low product yields [1,2]. Due to the drawbacks of chemical synthesis routes, alternative methods have been developed. Enzymatic synthesis is environmentally more friendly, highly selective, and efficient [3]. The most commonly applied enzyme-catalyzed reaction for the preparation of nucleosides and their analogues is the one-pot transglycosylation with pyrimidine and purine nucleoside phosphorylases [4]. Here, the sugar moiety is exchanged between two nucleobases in the presence of inorganic phosphate.

So far, mainly nucleoside phosphorylases (NP) of mesophilic organisms have been used for the synthesis of nucleoside analogues. However, enzymes from thermophilic bacteria or archaea are of increasing interest, as they are active over a wide temperature and pH range, which facilitates the solubility of substrates. Furthermore, mass transfer is improved due to a decreased viscosity and an increase in diffusion coefficient and flux at higher temperatures [5,6]. In addition, due to the ease of their purification, thermophilic enzymes are cost-effective biocatalysts for industrial processes [6].

Nucleoside analogues form a valuable class of drugs that have been widely used for more than 50 years for several indications such as cancer, viral, or protozoal infections [7,8,9]. Purine nucleoside analogues such as cladribine, fludarabine, or clofarabine are marketed drugs for the treatment of hematological malignancies [8]. These compounds are derivatives of adenosine and show modifications in the base and/or the sugar moiety. Cladribine, fludarabine, and clofarabine have a halogen atom at position 2 of the nucleobase, which was shown to be an important prerequisite for deaminase resistance and, therefore, its increased intracellular activity [10]. These compounds all share a similar mechanism of action, such as the inhibition of DNA synthesis, inhibition of DNA repair, and accumulation of DNA strand breaks.

Biological properties of enzymatically produced nucleoside analogues have been studied before. Analogues of 2′-deoxyribosides were produced using mesophilic nucleoside deoxyribosyltransferase from *Lactobacillus leichmannii* [11,12]. In a first attempt, 12 analogues of 2′-deoxyadenosine were enzymatically produced at a 100 to 400 mg scale with an average yield of 64% [11]. In a second study, 8-substituted purine nucleoside analogues were synthesized with yields of <10 to 70% [12]. Of the compounds produced, 2-chloro and 2-bromo analogues of 2′-deoxyadenosine were good inhibitors in tumor cell lines [11,12].

In the present study, the enzymatic synthesis of 2,6-dichloropurine riboside (**3a**), 2,6-dichloropurine deoxyriboside (**3b**), 6-chloro-2-fluoropurine riboside (**3c**), and 6-chloro-2-fluoropurine deoxyriboside (**3d**) with thermostable nucleoside phosphorylases was optimized based on thermodynamic equilibrium state calculations [13]. Equilibrium constants for dihalogenated nucleosides were calculated and used to determine optimum reactions to reach 90% or 95% product yields. The transferability of the results obtained on an analytical scale to up-scaling experiments was analyzed. Furthermore, we evaluated the cytotoxic activity of dihalogenated nucleoside analogues in a hematologic tumor cell line. While 2-halogenated nucleoside analogues were shown to have an increased stability and activity due to a resistance to deaminases [10], dihalogenated compounds have not been studied before.

## 2. Results

### 2.1. Optimization of the Synthesis of Dihalogenated Nucleosides Based on Thermodynamic Calculations

Transglycosylation reactions were employed to produce 2,6-dichloropurine (**2a**) and 6-chloro-2-fluoropurine (**2b**) containing nucleosides using uridine (**1a**) and thymidine (**1b**) as sugar donors (Scheme 1). Pyrimidine nucleoside phosphorylase (PyNP, EC 2.4.2.2) and purine nucleoside phosphorylase (PNP, EC 2.4.2.1) from a thermophilic organism were applied as biocatalysts.

To efficiently produce **2a**- and **2b**-containing nucleosides in a transglycosylation reaction, equilibrium state thermodynamic calculations were carried out [13]. We have recently shown that nucleoside phosphorolysis is a reversible endothermic reaction under tight thermodynamic control [14]. Consequently, it is possible to predict the equilibrium states of transglycosylation reactions, if the equilibrium constants of phosphorolysis of both participating nucleosides are known [13]. However, for **3a**–**d**, equilibrium constants of phosphorolysis have not been described before. Thus, the equilibrium states of analytical-scale reactions were used to calculate these values for **3a**–**d**, employing previously published equations (for more detail, please see reference [14], the respective Appendix A [15,16], as well as the externally hosted supplementary material of this publication [17]). The recently described equilibrium constants of the sugar donors **1a** and **1b** [14] served as the basis for these calculations. Analytical-scale reactions were performed with equal concentrations of base and sugar donor of 5 mM and 2 mM of phosphate. At equilibrium, product formation was between 55% (**3b**) and 60% (**3a** and **3c**).

Thus, the calculated equilibrium constants (K_2_) were in the range of 0.071 (**3d**) to 0.081 (**3a**) (Table 1). These values fit well with previously reported equilibrium constants of the phosphorolysis of other purine nucleosides [14]. Equilibrium constants were slightly higher for **2a**-derived nucleosides (**3a** and **3b**) compared to **2b**-derived nucleosides (**3c** and **3d**) and for deoxyribonucleosides in comparison to ribonucleosides (Table 1).

Employing these equilibrium constants, optimal reaction conditions to provide 90% or 95% conversion were calculated. Sugar donor excess was predicted to be in the range of 4.4 to 6.1 to reach 90% product yield and 8.8 to 12.5 to obtain 95% product formation (Table 1). In accordance with the equilibrium constants of **1a** and **1b**, a higher sugar donor excess was estimated to be needed for the synthesis of deoxyribonucleosides compared to the corresponding ribonucleosides.

Analytical-scale experiments with the selected conditions confirmed the theoretical predictions. Product formation of 90% and 95% was observed under the tested reaction conditions for all target compounds (Figure 1). However, the time to reach equilibrium differed between ribonucleosides and deoxyribonucleosides. While equilibrium was reached within 1 h for **3b** and **3d**, equilibrium was only observed after 6 h for **3a** and **3c** (Figure 1). Therefore, for semi-preparative-scale, reactions with ribonucleosides and deoxyribonucleosides were stopped after 6 h and 2 h, respectively. In semi-preparative experiments, conditions were used to reach 90% product yield. Doubling of the amount of **1a** and **1b** would have been needed to reach 95% product formation. However, high quantities of the sugar donor would increase the substrate cost and the ecological burden, as a lot of sugar donor is wasted. In addition, it complicates purification, as an efficient product separation is no longer ensured.

### 2.2. Synthesis of Dihalogenated Nucleosides in Semi-Preparative Scale

Dihalogenated nucleosides were produced at 50-mL scale to validate the efficiency and specificity of the synthesis and purification process. Products were purified by HPLC. Recovery of the sugar donor was validated for **3b** as an example. Appropriate reaction conditions were employed to reach 90% product yield.

Enzymatic transglycosylation reactions led to conversions of 83% (**3a**), 92% (**3b**), 90% (**3c**), and 92% (**3d**) (Figure 1, Appendix A). This was in good accordance with small-scale experiments. The time course of product formation was also well comparable with small-scale studies. Equilibrium was reached within 1 h for **3b** and **3d** and mainly within 6 h for **3a** and **3c** (Figure 1). Except for the expected reaction products uracil (**4a**) or thymine (**4b**), no further by-products were observed by HPLC analysis in any of the reactions.

Products synthesized in transglycosylation reactions were purified by semi-preparative HPLC. Proteins were removed by filtration as a prerequisite for this step. Product loss during protein removal and HPLC purification was in the range of 18% (**3a**) to 31% (**3b** and **3d**) (Appendix A). Still, this step was required to guarantee high product purity. Freeze-drying yielded the target compounds as white powders. The highest losses in product quantity (28% to 32%) were observed in this step (Appendix A). Losses can be reduced by collecting more of the product during HPLC purification and by conducting a second round of freeze-drying.

In total, 34 mg, 36 mg, 32 mg, and 32 mg of purified powder were obtained for **3a**, **3b**, **3c**, and **3d**, respectively (Appendix A). These values correspond to a recovery of approximately 50% for each target compound (Appendix A). Product purity was greater than 96%, and structures were confirmed by NMR (Appendix A, Appendix A and Appendix A).

Since the sugar donors used can be a cost factor for the enzymatic synthesis of nucleoside analogues, the recovery of **1b** was analyzed using the synthesis of **3b** as an example. In the enzymatic reaction, 369 mg of **1b** was added. The transglycosylation reaction resulted in 74 mg (20%) conversion to **4b**. After protein removal and HPLC purification, 251 mg of the remaining **1b** were recovered. In the freeze-drying process 22% of **1b** was lost and in total, 195 mg **1b** (53% yield) was recaptured from the initial 369 mg added as the starting material.

### 2.3. Effect of Dihalogenated Nucleoside Analogues on Cell Growth

Dihalogenated nucleosides **3a**–**d** were tested for their ability to inhibit the growth of HL-60 cells in culture. Monohalogenated cladribine, which is known to efficiently inhibit HL-60 cells [18,19], was used as positive control. While for cladribine an IC_50_ value of 0.22 µM [± 0.11 µM] was determined, 50% of inhibition by the dihalogenated nucleoside analogues was observed for concentrations between 10 and 100 µM (Figure 2). The inhibition of growth of the control cell line HEK293 was in the similar range of nucleoside analogue concentrations. No unspecific growth inhibition in the control cell line HEK293 was detected for cladribine. Hence, nucleoside analogues with halogen atoms in positions 2 and 6 are no suitable inhibitors of leukemia cell lines.

## 3. Discussion

The current study shows the feasibility of optimizing the enzymatic synthesis of modified nucleosides based on initial experiments and thermodynamic calculations to determine equilibrium constants. Dihalogenated nucleoside analogues **3a**–**d** were formed with percentages of conversion of 90% to 95% at small scale and 83% to 92% at semi-preparative scale. Yields of purified product were approximately 50%.

Dihalogenated purine bases have been studied before as substrates for NPs from extreme environments. Both **2a** and **2b** were converted to the respective riboside and deoxyriboside nucleosides by a thermostable PNP from *Geobacillus thermoglucosidasius* with conversions of 54% to 69% [20]. A sugar donor-to-nucleobase ratio of 3.34 in the presence of 3.34 equivalents of phosphate (compared to the base concentration) was used. Using an immobilized PyNP of *Thermus thermophilus* and the same PNP, Zhou and colleagues later observed increasing product formation with reduced phosphate concentrations [21]. A negative impact on the product yield of phosphate equivalents above 0.3 in comparison to the nucleobase was recently confirmed by thermodynamic studies on transglycosylation reactions [13,22]. The advantages of the application of thermodynamic equilibrium state calculations in transglycosylation reactions is convincingly confirmed in the present study, which also represents the first example of the fruitful application of this methodology in practice. After calculating the equilibrium constants based on a single analytical reaction each, reaction conditions were designed to reach product yields of 90% and 95%. Small- and semi-preparative-scale experiments employing the improved conditions confirmed the calculated predictions. Hence, under optimized conditions, higher product yields were observed compared to previous studies [20,21].

Besides the optimization of the enzymatic synthesis process by thermodynamic calculation, the present study also investigated the impact of a chlorine atom at position 6 of the nucleoside with respect to its inhibitory effect on leukemic cells. This was based on the observation that halogenation is an important prerequisite for anticancer activity [3,23,24], since amino groups make the compound susceptible to deamination by adenosine deaminase (ADA) [25,26]. The addition of a chlorine or fluorine atom in position 2 of adenosine strongly increased its stability, as observed for the very potent leukemia drugs cladribine and fludarabine [23,26]. Interestingly, the dihalogenated analogues tested in the present study with cell line HL-60 showed less growth inhibition than the monohalogenated cladribine. An explanation for the low activity could involve the uptake of the dihalogenated nucleosides, the activation by nucleoside and nucleotide kinases, or the interaction with the intracellular targets. The uptake of cladribine involves equilibrative nucleoside transporter 1 and 2 as well as concentrative nucleoside transporter 3 [27], which might not recognize dihalogenated nucleosides. Most of the known nucleoside analogue drugs including cladribine are only active as the respective di- or triphosphorylated nucleotides, and activation in vivo is often insufficient [28]. Hence, a lack of activity might be explained by human kinases not efficiently phosphorylating dihalogenated nucleosides. Intracellular targets of cladribine have been recently well summarized by Freyer and colleagues [29]. Ribonucleotide reductase is one of the main targets of cladribine. Its inhibition leads to a depletion of deoxyadenosine and hence to an imbalance within the deoxyribonucleotide pool. This leads to a number of other effects such as endonuclease activation and double-strand DNA breaks. Additionally, the inhibition of adenosine deaminase [30] or RNA polymerase [31] by cladribine were described before. However, further studies are needed to better understand the limitations of dihalogenated nucleosides in the inhibition of leukemia cell lines.

Although the dihalogenated nucleoside analogues showed less growth inhibition in leukemia cell line HL-60 than their monohalogenated counterparts, they may also serve as valuable starting material for the preparation of a broad palette of modified nucleosides [32,33]. Dihalogenated nucleoside analogues have a much higher solubility than monohalogenated analogues [20] and can therefore serve as precursors for the synthesis of new compounds or approved substances in industrial scale.

## 4. Materials and Methods 

### 4.1. General Information

All chemicals and solvents were of analytical grade or higher and purchased, if not stated otherwise, from Sigma-Aldrich (Steinheim, Germany), Carl Roth (Karlsruhe, Germany), TCI Deutschland (Eschborn, Germany), and VWR (Darmstadt, Germany). Cladribine was obtained from TCI with a purity >98%. Water was purified and deionized by an EASYpure II purification system to reach a resistivity of 18.2 MΩ cm at 25 °C (Werner, Leverkusen, Germany). Thermostable nucleoside phosphorylases (PyNP 02 [E-PyNP-0002] and PNP 02 [E-PNP-0002]) were provided by BioNukleo and used as recommended by the manufacturer. Nucleoside phosphorylases were heterologously expressed in *E. coli* and originate from thermophilic bacteria with temperature optima at 60 °C. Enzymes were purified by affinity chromatography. Enzyme concentrations of 0.1 mg mL^−1^ were used for PyNP and PNP. PyNP 02 and PNP 02 had activities of 40 U mL^−1^ and 60 U mL^−1^ at 40 °C and pH 9 for their natural substrates uridine and adenosine, respectively, as determined by the UV spectroscopy-based assay described recently [34]. Cell lines HEK-239 and HL-60 were purchased from the German Collection of Microorganisms and Cell Cultures GmbH. All experimental and calculated data are available free of charge from an external online repository [17].

### 4.2. Optimization of the Enzymatic Synthesis of Dihalogenated Purine Nucleosides

Thermodynamic calculations were shown to allow for a reliable calculation of product yields in nucleoside phosphorylase catalyzed transglycosylation reactions based on equilibrium constants of phosphorolysis [13,22]. Equilibrium constants for the phosphorolysis of uridine (**1a**) and thymidine (**1b**) were previously described to be 0.18 and 0.125, respectively, at 40 °C [14]. To estimate equilibrium constants K_2_ for the synthesis of the dihalogenated nucleoside analogues, enzymatic reactions were performed with 5 mM of sugar donor (**1a** or **1b**), 5 mM of the purine base **2a** or **2b**) in 2 mM potassium phosphate (pH 7.5) using 0.1 mg mL^−1^ of the biocatalysts PyNP 02 and PNP 02. Reactions were performed at 40 °C for 20 h. To analyze sugar donor cleavage and product formation, samples were analyzed by HPLC. First, 30 µL were removed from the reaction mixture, and an equal volume of methanol was added to stop the reaction. After centrifugation at 21,500 g for 10 min, the supernatant was diluted 1:10 and transferred to HPLC vials.

Solving the system of equilibrium constraints given by the laws of mass action of the first and second (reverse) phosphorolysis of the transglycosylation sequence for these values, as described recently [13,14], yielded equilibrium constants of phosphorolysis for **3a**–**d**. Based on these equilibrium constants, the sugar donor excess needed to reach 90% and 95% product formation was calculated [13]. Sugar donor excess was calculated to be between 4.4 and 6.1 to reach 90% product yield and 8.8 and 12.5 to guarantee 95% product formation (Appendix A). Reactions were performed to verify theoretical calculations at 1 mL scale. Base concentrations of 5 mM and enzyme concentration of 0.1 mg mL^−1^ for PyNP 02 and PNP 02 were used in 0.5 mM potassium phosphate buffer (pH 7.5). The reaction temperature was 40 °C. Regular samples were taken to analyze sugar donor cleavage and product formation by HPLC. A 30 µL sample was removed from the reaction mixture, and an equal volume of methanol was added to stop the reaction. After centrifugation at 21,500 g for 10 min, the supernatant was diluted 1:10 and transferred to HPLC vials.

### 4.3. Enzymatic Production of Dihalogenated Purine Nucleosides

To produce dihalogenated nucleoside analogues, base concentrations of 5 mM and sugar donor concentrations of 22 to 30.5 mM were applied in a reaction volume of 50 mL. Thermostable pyrimidine and purine nucleoside phosphorylase PyNP 02 and PNP 02 were added in a concentration of 0.1 mg mL^−1^ each. Enzymatic reactions were performed in 0.5 mM potassium phosphate buffer at pH 7.5 using a reaction temperature of 40 °C. Reactions were stirred at 100 rpm. Regular samples were taken to monitor enzymatic reactions by HPLC as described above.

The reactions were stopped by enzyme removal after 2 h and 6 h for deoxyriboside and riboside nucleosides, respectively. Protein was removed by filtration using centrifugal filter units with a cut-off of 30 kDa (Millipore, Darmstadt, Germany) at room temperature. Filtrate was stored at 4 °C until purification by semi-preparative HPLC.

Analytical HPLC was performed to determine product quantity after each step. Product loss per step was determined based on the product quantity of step 1 (e.g., after enzymatic synthesis) and the product quantity of the following step (e.g., after protein depletion and HPLC).

### 4.4. Cell Culture

HEK293 cells were cultured in low-glucose Dulbecco’s modified Eagle’s medium (Biowest) containing 10% fetal bovine serum, 2 mM glutamine, non-essential amino acids, 4.5 mg mL^−1^ glucose, and the antibiotics penicillin and streptomycin. HL-60 cells were cultured in RPMI 1640 w/o l-Glutamine supplemented with 10% fetal bovine serum, 2 mM glutamine, and the antibiotics penicillin and streptomycin. Cells were grown at 37 °C in a humidified atmosphere with 5% CO_2_.

### 4.5. XTT Cell Proliferation Assay

Cytotoxicity was measured by the XTT assay at 450 nm as described by the manufacturer (Roche Diagnostics, Basel, Switzerland). Phosphate-buffered saline was used as negative control. For HL-60 and HEK293 cells, 1.2 × 10^5^ cells in growth media were transferred to each well of a 96-well flat-bottom plate. Subsequently, modified nucleosides were added in varying concentrations. XTT assay was performed after 24 h for HL-60 and HEK293 cells, as described previously [35]. At least three replicates for each nucleoside concentration were performed. All experiments were conducted for a minimum of three independent experiments.

### 4.6. Analytical HPLC

Nucleosides and bases were analyzed by HPLC (λ = 260 nm) using a reversed-phase Kinetex EVO C18 column (Phenomenex, Aschaffenburg, Germany) at 25 °C with a flow rate of 1 mL min^−1^. Substrates and products were identified by comparing their retention times with those of authentic standards. Quantification was performed using a six-step calibration from a serial dilution in the range from 0 to 1 mM over the peak area.

Gradual elution was performed using eluent A (20 mM ammonium acetate) and eluent B (100% acetonitrile). The following gradient was used: 97% A and 3% B; 10-min linear gradient to 60% A and 40% B. The initial condition (97% A and 3% B) was restored afterwards and maintained for 4 min.

Typical retention times were as follows: **1a**: 3.2 min, **4a**: 2.4 min, **1b**: 4.7 min, **4b**: 4 min, **3c**: 7.5 min, **3d**: 8.2 min, **2b**: 7 min, **3a**: 8.2 min, **3b**: 8.8 min, and **2a**: 7.6 min. The percentage of conversion of nucleosides was calculated as described previously (Equation (1)) [36].
(1)$$\mathrm{Conversion}\;\%=\frac{\mathrm{Conc}.\;\mathrm{of}\;\mathrm{the}\;\mathrm{product}\;\left[\mathrm{mM}\right]}{\mathrm{Conc}.\;\mathrm{of}\;\mathrm{the}\;\mathrm{product}\;\left[\mathrm{mM}\right]+\;\mathrm{Conc}.\;\mathrm{of}\;\mathrm{the}\;\mathrm{substrate}\;\left[\mathrm{mM}\right]}\times \;100$$

### 4.7. Purification of Nucleoside Analogues by Semi-Preparative HPLC

Nucleoside analogues were purified at room temperature using a KNAUER HPLC system equipped with a Smartline Detector 2600 and an AZURA P 2.1 L pump. A flow rate of 21.24 mL min^−1^ and a Kinetex^®^ 5 μm EVO C18 250 × 21.2 mm RP column were used. Deionized water and acetonitrile were used as eluents, while the gradient was modified from the analytical method as summarized in Appendix A. Fractions containing either the sugar donor or the product were collected.

Collected samples were dried using a Christ Gamma 1–20 freeze-dryer (Osterode am Harz, Germany). Residual water was removed by drying at 50 °C in a drying cabinet (DRY-Line 112 Prime, VWR) for 8 h.

### 4.8. NMR Analysis

The NMR spectra for ^1^H, ^13^C, and ^19^F were recorded in DMSO-*d*_6_ (purchased from *Eurisotop*) on a Bruker Avance III 500 MHz instrument. Chemical shifts are reported in parts per million (ppm) and are referenced to the residual solvent resonance as the internal standard (^1^H NMR: δ = 2.50 ppm for DMSO-*d*_5_; ^13^C NMR: δ = 39.52 ppm for DMSO-*d*_6_) [37]. ^19^F NMR spectra are referenced in compliance with the unified scale for NMR chemical shifts as recommended by the IUPAC stating the chemical shift relative to CCl_3_F [38]_._ Data are reported as follows: chemical shift, multiplicity (s = singlet, d = doublet, m_c_ = centrosymmetric multiplet, br = broad signal), coupling constants (Hz), and integration.

## 5. Conclusions

The present study demonstrates the value of small-scale experiments combined with thermodynamic equilibrium state calculations to predict conditions for the efficient synthesis of nucleosides at a larger scale. This approach was used to produce **3a**–**d** in an enzymatic transglycosylation reaction. Upscaling of the reaction was successfully performed based on the theoretical predictions. The presented strategy can be transferred to the synthesis of a wide range of nucleoside analogues using nucleoside phosphorylases as biocatalysts.

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
