# Peer review of "Efficient Biocatalytic Synthesis of Dihalogenated Purine Nucleoside Analogues Applying Thermodynamic Calculations"

_molecules, 2020, doi:10.3390/molecules25040934_

Round 1
Reviewer 1 Report
The manuscript presented by Heba et co-workers aims to offer the integration of biocatalysis and thermodynamic calculations to design more efficient bioprocesses. However in the last 2 years it has been published up to 4 articles which looks like very similar than the present. In this respect, despite the present manuscript includes a lot of experimental work, the novelty is low and the therapeutic effect of synthesized nucleosides is very poor.
Major points:
A) The synthesis of these type of 2,6 halogenated nucleoside by nucleoside phosphorylases is no new, and was reported by the the same authors.
-Zhou, X., Szeker, K., Jiao, L. Y., Oestreich, M., Mikhailopulo, I. A., & Neubauer, P. (2015). Synthesis of 2, 6‐dihalogenated purine nucleosides by thermostable nucleoside phosphorylases. Advanced Synthesis & Catalysis, 357(6), 1237-1244.
B) The same hypothesis about the thermodynamic calculations has been employed by the authors very recently, using also the same enzymes.
-Kaspar, F., Giessmann, R. T., Neubauer, P., Wagner, A., & Gimpel, M. Thermodynamic Reaction Control of Nucleoside Phosphorolysis. Advanced Synthesis & Catalysis.(2019).
C) Moreover these enzymes has been also used in other manuscript in a similar way:
-Kaspar, F., Giessmann, R. T., Hellendahl, K., Neubauer, P., Wagner, A., & Gimpel, M. (2019). General Principles for Yield Optimization of Nucleoside Phosphorylase‐Catalyzed Transglycosylations. ChemBioChem.
D) In addition, a similar approach (without thermodynamic calculations) has been described for othe nucleoside phosphorylases
-Alexeev, C. S., Kulikova, I. V., Gavryushov, S., Tararov, V. I., & Mikhailov, S. N. (2018). Quantitative Prediction of Yield in Transglycosylation Reaction Catalyzed by Nucleoside Phosphorylases. Advanced Synthesis & Catalysis, 360(16), 3090-3096.
-Giessmann, R. T., Krausch, N., Kaspar, F., Cruz Bournazou, M. N., Wagner, A., Neubauer, P., & Gimpel, M. (2019). Dynamic Modelling of Phosphorolytic Cleavage Catalyzed by Pyrimidine-Nucleoside Phosphorylase. Processes, 7(6), 380.
E) Finally the therapeutic effect of the nucleosides could be the novelty of this work, but IC50 values are very low, so these compounds are not suitable inhibitors of leukemia
Minor points:
- Authors must discuss why the therapeutic effect of the nucleosides is so low, including information about what should be the therapeutic targets affected by these nucleosides.
Author Response
The manuscript presented by Heba et co-workers aims to offer the integration of biocatalysis and thermodynamic calculations to design more efficient bioprocesses. However in the last 2 years it has been published up to 4 articles which looks like very similar than the present. In this respect, despite the present manuscript includes a lot of experimental work, the novelty is low and the therapeutic effect of synthesized nucleosides is very poor.
Thank you for your input. The comments have helped to further improve the significance of our manuscript. We hope that our answers and additional information in the manuscript will help to highlight the novelty of the work more clearly.
The presented manuscript is focusing on the following topics, which have not been presented before:
- Are thermodynamic calculations applicable for dihalogenated nucleosides? Specifically, do transglycosylation reactions with these nucleosides behave according to equilibrium constraints observed for other systems?
- Can equilibrium constants be determined from just single analytical scale transglycosylation experiment?
- Do optimized reaction conditions based on thermodynamic calculations lead to higher product yields compared to earlier published data?
- Are results based on thermodynamic calculations transferable from analytical scale to larger scale?
- Do dihalogenated nucleosides show growth inhibitory activity in leukemia cell lines?
The manuscripts focusing on thermodynamic studies published within the last 2 years laid the groundwork for the present study.
Further information was added to the manuscript to make the novelty of our study clearer.
Major points:
A) The synthesis of these type of 2,6 halogenated nucleoside by nucleoside phosphorylases is no new, and was reported by the the same authors.
We agree that dihalogenated compounds were studied before as substrates for nucleoside phosphorylases. However, in comparison to previous studies the advantages of thermodynamic calculations are clearly shown. Product yields obtained after rational optimization of the reaction conditions were higher compared to previous studies. Thus, we are presenting a significantly improved synthesis of these compounds, thereby demonstrating the first practical example of how equilibrium state thermodynamics can be employed as a tool in biocatalytic nucleoside synthesis. In addition, the reactions were carried out in a larger scale, demonstrating scalability of the previously described systems.
B) The same hypothesis about the thermodynamic calculations has been employed by the authors very recently, using also the same enzymes.
-Kaspar, F., Giessmann, R. T., Neubauer, P., Wagner, A., & Gimpel, M. Thermodynamic Reaction Control of Nucleoside Phosphorolysis. Advanced Synthesis & Catalysis.(2019).
The fundamental studies of Kaspar et al. (ASC and chembiochem) served as basis for the recent publication. However, as stated above the applicability of thermodynamic calculations for up-scaled reactions is the focus of the recent manuscript. In addition to previous work, we demonstrated herein that thermodynamics-guided reaction optimization can be carried out without prior knowledge of the equilibrium constants of the product nucleoside and without access the product nucleoside.
C) Moreover these enzymes has been also used in other manuscript in a similar way:
-Kaspar, F., Giessmann, R. T., Hellendahl, K., Neubauer, P., Wagner, A., & Gimpel, M. (2019). General Principles for Yield Optimization of Nucleoside Phosphorylase‐Catalyzed Transglycosylations. ChemBioChem.
We agree that the applied enzymes were used in a previous study. However, they proved suitable tools to produce dihalogenated purine nucleosides. As discussed in our previous work (Kaspar et al. 2019, ASC), phosphorolysis and transglycosylation yields are independent of the enzymes used for catalysis. Thus, it was not necessary to employ other/novel enzymes for the present synthetic task.
D) In addition, a similar approach (without thermodynamic calculations) has been described for other nucleoside phosphorylases
-Alexeev, C. S., Kulikova, I. V., Gavryushov, S., Tararov, V. I., & Mikhailov, S. N. (2018). Quantitative Prediction of Yield in Transglycosylation Reaction Catalyzed by Nucleoside Phosphorylases. Advanced Synthesis & Catalysis, 360(16), 3090-3096.
-Giessmann, R. T., Krausch, N., Kaspar, F., Cruz Bournazou, M. N., Wagner, A., Neubauer, P., & Gimpel, M. (2019). Dynamic Modelling of Phosphorolytic Cleavage Catalyzed by Pyrimidine-Nucleoside Phosphorylase. Processes, 7(6), 380.
The publications of Alexeev et al. and Giessmann et al. were important publications showing the possibility to use thermodynamic calculations in NP-catalyzed phosphorolysis and transglycosylation reactions. Formulas were developed to calculate maximum product yields. Both publications served as basis for the recent study as we applied these formulas to produce interesting nucleoside analogues in an enzymatic process.
E) Finally the therapeutic effect of the nucleosides could be the novelty of this work, but IC50 values are very low, so these compounds are not suitable inhibitors of leukemia
We agree that the therapeutic potential is lower for the dihalogenated compounds compared to cladribine. However, in our opinion this is also a valuable information, which opens space for further investigations and consequently should be communicated to the scientific community. Although dihalogenated compounds are no suitable compounds to treat leukemia they are still highly interesting scaffolds in their own right, e.g. due to their high solubility. Hence, they could be used as precursors for the synthesis of compounds like cladribine.
Minor points:
- Authors must discuss why the therapeutic effect of the nucleosides is so low, including information about what should be the therapeutic targets affected by these nucleosides.
Thank you for raising this very interesting point. We have included more information in the manuscript regarding the mechanism of action of cladribine (on lines 226-238).
Reviewer 2 Report
The paper by Yehia et al. describes the enzymatic synthesis of 2,6-dihalogenated purine ribo- and 2’-deoxyribonucleosides and the biological evaluation of the synthesized molecules as potential anticancer agents. The biological assay was performed against leukemic cell line HL-60 by using cladribine, an established anti-leukaemia drug, as the reference standard.
The synthesis of the target molecules was achieved by using two thermostable nucleoside phosphorylases, previously produced and investigated by some of the authors in the synthesis of nucleoside analogues (see refs. 18-19). Experimentals were supported by theoretical calculations aimed at finding the optimal reagent ratio for the highest yield, according to a previous report describing the optimization of the enzymatic synthesis of both natural nucleosides and a number of modified nucleosides by transglycosylation (see refs. 14-15).
Since the highest yield was achieved by using a large excess of the sugar donor (e.g. Thd), the unreacted nucleoside was conveniently recovered (53% yield) in the preparative synthesis of DCP-dR.
The paper describes synthesis, characterization, and biological evaluation of new molecules by using an efficient enzymatic route and a rational approach for the experimental design. I think that the paper can be accepted for publication in Molecules, although not in the current form. There are in fact some issues that, in this Reviewer’s opinion, needs to be better explained and described.
As a general comment, the Authors tend to overestimate the knowledge of the reader about the state-of-the-art that this paper moves from and within. I suggest to explain more in detail some concepts and previous data (see below). Although the work is overall clear and the results worth of publication, description of the main goal of the research is not as clear. On one hand, validation of previous thermodynamics calculations in the synthesis of dihalogenated purine nucleosides appears to be the aim of work, on the other hand, discussion about the biological activity of the newly synthesized compounds diverts the attention from that. I think that Authors should better balance these two contributions.
Abstract
33: cladribine is an established drug; thus, it does not show a “potential” but a high antiproliferative “activity”.
A brief explanation/more details on the thermodynamics calculations (see the manuscript title) should be added in the abstract.
Suggestion: for a better focus on the aim of work, concepts reported in lines 20-22 could be moved at the end of the abstract (before “Taken together…”) and suitably integrated with the current text.
Introduction
63: why mass transfer is improved?
71-73: this sentence is misleading. As far as the sentence is reported, it seems that a difference can be found when the same molecule is chemically or enzymatically synthesized…What it counts is the identity of the molecule, not the route to obtain it. Please, re-phrase and/or explain this concept.
81-85: this paragraph describes the results. It should be moved from the Introduction to a different section of the paper.
Suggestion: for an easier reading, focus the first section on the enzymatic synthesis and the second section on biological activity and structure-activity relationships.
Results, Discussion and artwork
Scheme 1 and text: I warmly suggest to number each molecule and to report the structure legenda as follows:
DCP-R: R2=OH, R3=Cl, R4=Cl and so on
For example, Urd (1) and Thd (2), DCP (3), CFP (4), DCP-R (5) and so on. Alternatively, DCP-R could be numbered 3a and DCP-dR 3b…and so on. Moreover, the by-products Ura and Thy should be moved at the end of the scheme and numbered in a consecutive mode (for instance, 9 and 10 or 5 and 6, depending on the target molecules numbering)
90-91: the biological source of the enzymes (i.e. thermophilic microorganism and EC number) should be included.
100-106: for a full comprehension, this paragraph requires that the reader has read refs. 14 and 15. A short sentence which summarizes the rationale and the main findings of those previous works is necessary (either here or in the Discussion section).
157: this Reviewer would expect that uracil and/or thymine is formed as a by-product during the reaction course (according to Scheme 1). So, the sentence “No byproducts were observed in any of the reactions by HPLC” should be explained.
160: “HPLC protocols were optimized for each target compound to guarantee efficient product purification.” This is an obvious sentence. I suggest to delete it.
158-164: purification step and freeze drying procedure appeared to be the sore point affecting the product final yield (an average of 50% yield is obtained from a 90-95% conversion). How the authors think that the process could be made more efficient?
Both in the text and in Table 2 “protein depletion” should be replaced with “protein removal”, “protein filtration”, “protein downstream”, or something like that.
188: “validate” does not seem to be an appropriate terminology (performed).
204-205: “The current study shows the feasibility of optimizing the enzymatic synthesis of modified nucleosides using thermodynamic calculations”. This sentence is misleading. Optimization of the enzymatic synthesis of nucleosides is the result of previous investigations (refs. 14-15). I suggest to re-phrase this sentence accordingly.
References
References must be double-checked and edited by following the Journal guidelines.
Author Response
The paper by Yehia et al. describes the enzymatic synthesis of 2,6-dihalogenated purine ribo- and 2’-deoxyribonucleosides and the biological evaluation of the synthesized molecules as potential anticancer agents. The biological assay was performed against leukemic cell line HL-60 by using cladribine, an established anti-leukaemia drug, as the reference standard.
The synthesis of the target molecules was achieved by using two thermostable nucleoside phosphorylases, previously produced and investigated by some of the authors in the synthesis of nucleoside analogues (see refs. 18-19). Experimentals were supported by theoretical calculations aimed at finding the optimal reagent ratio for the highest yield, according to a previous report describing the optimization of the enzymatic synthesis of both natural nucleosides and a number of modified nucleosides by transglycosylation (see refs. 14-15).
Since the highest yield was achieved by using a large excess of the sugar donor (e.g. Thd), the unreacted nucleoside was conveniently recovered (53% yield) in the preparative synthesis of DCP-dR.
The paper describes synthesis, characterization, and biological evaluation of new molecules by using an efficient enzymatic route and a rational approach for the experimental design. I think that the paper can be accepted for publication in Molecules, although not in the current form. There are in fact some issues that, in this Reviewer’s opinion, needs to be better explained and described.
As a general comment, the Authors tend to overestimate the knowledge of the reader about the state-of-the-art that this paper moves from and within. I suggest to explain more in detail some concepts and previous data (see below).
Although the work is overall clear and the results worth of publication, description of the main goal of the research is not as clear. On one hand, validation of previous thermodynamics calculations in the synthesis of dihalogenated purine nucleosides appears to be the aim of work, on the other hand, discussion about the biological activity of the newly synthesized compounds diverts the attention from that. I think that Authors should better balance these two contributions.
Many thanks for the very interesting comments and suggestions. We have implemented these points and agree that the manuscript has gained in value once again.
Abstract
33: cladribine is an established drug; thus, it does not show a “potential” but a high antiproliferative “activity”.
The text was changed as suggested.
A brief explanation/more details on the thermodynamics calculations (see the manuscript title) should be added in the abstract.
We added a brief statement to the abstract and significantly expanded the respective paragraph in the results section (please see below).
Suggestion: for a better focus on the aim of work, concepts reported in lines 20-22 could be moved at the end of the abstract (before “Taken together…”) and suitably integrated with the current text.
The text was changed as suggested.
Introduction
63: why mass transfer is improved?
Increase in temperature leads to an increase in diffusion coefficient and flux. Furthermore, the apparent viscosity is decreased with an increase in temperature which leads to an improved mass transfer. Further information was added to the text (line 53-54).
71-73: this sentence is misleading. As far as the sentence is reported, it seems that a difference can be found when the same molecule is chemically or enzymatically synthesized…What it counts is the identity of the molecule, not the route to obtain it. Please, re-phrase and/or explain this concept.
We agree that the sentence is misleading. As a comparison of enzymatically and chemically produced compounds is not the topic of the manuscript, we deleted the sentence.
81-85: this paragraph describes the results. It should be moved from the Introduction to a different section of the paper.
The text was changed as suggested.
Suggestion: for an easier reading, focus the first section on the enzymatic synthesis and the second section on biological activity and structure-activity relationships.
The introduction was re-structured as suggested.
Results, Discussion and artwork
Scheme 1 and text: I warmly suggest to number each molecule and to report the structure legenda as follows:
DCP-R: R2=OH, R3=Cl, R4=Cl and so on
For example, Urd (1) and Thd (2), DCP (3), CFP (4), DCP-R (5) and so on. Alternatively, DCP-R could be numbered 3a and DCP-dR 3b…and so on. Moreover, the by-products Ura and Thy should be moved at the end of the scheme and numbered in a consecutive mode (for instance, 9 and 10 or 5 and 6, depending on the target molecules numbering)
The text and scheme 1 were changed as suggested.
90-91: the biological source of the enzymes (i.e. thermophilic microorganism and EC number) should be included.
Further information was added to the text (lines 86 and 87).
100-106: for a full comprehension, this paragraph requires that the reader has read refs. 14 and 15. A short sentence which summarizes the rationale and the main findings of those previous works is necessary (either here or in the Discussion section).
Thank you for raising this point. We followed your suggestion and included a brief summary of recent publications and their role in the present work. Additionally, we expanded on the methodology pertaining the thermodynamic calculations and added references to previously published work and data for additional clarification. To further facilitate reproducibility and comprehensibility, we are going to upload all of our experimental and calculated data at a freely accessible external online repository.
157: this Reviewer would expect that uracil and/or thymine is formed as a by-product during the reaction course (according to Scheme 1). So, the sentence “No byproducts were observed in any of the reactions by HPLC” should be explained.
The sentence was re-phrased to “Except for the expected reaction products uracil or thymine no further byproducts were observed in any of the reactions by HPLC.” (lines 147 to 149).
160: “HPLC protocols were optimized for each target compound to guarantee efficient product purification.” This is an obvious sentence. I suggest to delete it.
The respective sentence was deleted.
158-164: purification step and freeze drying procedure appeared to be the sore point affecting the product final yield (an average of 50% yield is obtained from a 90-95% conversion). How the authors think that the process could be made more efficient?
Losses observed in the purification step can be reduced by collecting more of the target peak in semi-preparative HPLC. In this study, we decided to collect less of the product to avoid impurities. Losses detected while freeze-drying are mainly based the collection of the powder. Losses can be reduced by solving remaining powder and have a second round of freeze-drying. We added a respective sentence on lines 166 to 167.
Both in the text and in Table 2 “protein depletion” should be replaced with “protein removal”, “protein filtration”, “protein downstream”, or something like that.
“Protein depletion” was replaced by “protein removal”.
188: “validate” does not seem to be an appropriate terminology (performed).
“Validate” was replaced by “analyzed”.
204-205: “The current study shows the feasibility of optimizing the enzymatic synthesis of modified nucleosides using thermodynamic calculations”. This sentence is misleading. Optimization of the enzymatic synthesis of nucleosides is the result of previous investigations (refs. 14-15). I suggest to re-phrase this sentence accordingly.
The sentence was rephrased to: “The current study shows the feasibility of optimizing the enzymatic synthesis of modified nucleosides based on initial experiments and thermodynamic calculations to determine equilibrium constants.”
Previous investigations provided equilibrium constants for uridine/thymidine and formulas to calculate equilibrium constants for nucleosides and to calculate sugar donor excess to reach product yields of 90% or 95%.
References
References must be double-checked and edited by following the Journal guidelines.
References were checked and edited following the Journal guidelines.
Reviewer 3 Report
2020.2.7
Manuscript ID: molecules-720285
Type of manuscript: Article
Title: Efficient biocatalytic synthesis of dihalogenated purine nucleoside analogues applying thermodynamic calculations
Authors: Heba Yehia, Sarah Westarp, Viola Röhrs, Felix Kaspar, Robert Giessmann, Hendrik Klare, Katharina Paulick, Peter Neubauer, Jens Kurreck, Anke Wagner *
Dear Prof. Grace Zhang
Assistant Editor, MDPI
Comments to the author
The manuscript submitted by Wagner et al. describes the experiments in which enzymatic synthesis of unnatural halogenated purine nucleosides applying equilibrium state thermodynamic calculations. The research findings of this paper will be important for the production of purine nucleoside analogues as drugs by bio-catalysis. Taken together, this work is certainly suitable for publication in “Molecules”. However, there are a few points the authors should address before publication.
Table 1. The ratio “%” need not show in column at product formation [%] at equilibrium. Author should add the data plots of enzymatic reaction under the both 5 mM of concentration of nucleobase and sugar donor in Figure 1. Then, I think the figures would make clear the benefit of thermodynamic calculation. Table 2 and 3. Is these information important? Author should move these two tables to “Supplementary Materials”. Line 255. In this manuscript, author use the mg/mL of enzyme concentration. I think author should use units as enzyme activity.
Author Response
The manuscript submitted by Wagner et al. describes the experiments in which enzymatic synthesis of unnatural halogenated purine nucleosides applying equilibrium state thermodynamic calculations. The research findings of this paper will be important for the production of purine nucleoside analogues as drugs by bio-catalysis. Taken together, this work is certainly suitable for publication in “Molecules”. However, there are a few points the authors should address before publication.
Thank you very much for the helpful comments. They further improved our manuscript.
Table 1. The ratio “%” need not show in column at product formation [%] at equilibrium.
Table 1 was changed as suggested.
Author should add the data plots of enzymatic reaction under the both 5 mM of concentration of nucleobase and sugar donor in Figure 1. Then, I think the figures would make clear the benefit of thermodynamic calculation.
End-point studies were used to determine equilibrium constants (two consecutive measurement revealed the same conversion). Hence, it would not be appropriate to include this data in Figure 1.
Table 2 and 3. Is these information important? Author should move these two tables to “Supplementary Materials”.
We agree that Tables 2 and 3 are of less importance and we moved them to the “Supplementary material” as suggested.
Line 255. In this manuscript, author use the mg/mL of enzyme concentration. I think author should use units as enzyme activity.
Specific activities of the enzymes were added to the manuscript (lines 258 to 260).
Reviewer 4 Report
The present study demonstrates the efficient enzymatic synthesis (DCP-R, DCP-dR, CFP-R and CFP-dR) of nucleoside aided by the prediction of thermodynamic equilibrium state calculations. The reaction was successfully performed based on the theoretical predictions and experiments of enzymatic transglycosylation reactions in a small scale. Though the new nucleosides obtained showed week anticancer activity, but the strategy developed can be transferred to the synthesis of a wide range of nucleoside analogues. The paper is well organized and could be published after minor revision.
1) Line 113, “be in the range of 4.5 to 6.1” should be “be in the range of 4.4 to 6.1” based on Table 1.
2) Regarding to the expression of product loss during protein depletion and HPLC purification, it’s not clearly stated how to get loss amount of per step. In addition, the binding affinities of Thd and Thymine to enzyme could be different. Furthermore, in the calculation of conversion by analytic HPLC, the step of protein depletion is also performed, is there any loss happened?
Author Response
The present study demonstrates the efficient enzymatic synthesis (DCP-R, DCP-dR, CFP-R and CFP-dR) of nucleoside aided by the prediction of thermodynamic equilibrium state calculations. The reaction was successfully performed based on the theoretical predictions and experiments of enzymatic transglycosylation reactions in a small scale. Though the new nucleosides obtained showed week anticancer activity, but the strategy developed can be transferred to the synthesis of a wide range of nucleoside analogues. The paper is well organized and could be published after minor revision.
Thank you very much for your input. It helped to improve our manuscript.
1) Line 113, “be in the range of 4.5 to 6.1” should be “be in the range of 4.4 to 6.1” based on Table 1.
The text was changed as suggested.
2) Regarding to the expression of product loss during protein depletion and HPLC purification, it’s not clearly stated how to get loss amount of per step.
Analytical HPLC was performed to determine product quantity after each step to produce dihalogenated nucleosides. Product loss per step was determined based on the product quantity of step 1 (e.g. after enzymatic synthesis) and the product quantity of the following step (e.g. after protein depletion and HPLC).
Further text was added to the Materials and Methods section (lines: 299-302).
In addition, the binding affinities of Thd and Thymine to enzyme could be different.
We agree that the binding affinities of Thd and Thy could be different; however, this does not influence the thermodynamic equilibrium of the observed reactions.
Furthermore, in the calculation of conversion by analytic HPLC, the step of protein depletion is also performed, is there any loss happened?
We also determined losses due to protein removal using 3b as an example. Losses were below 1%. Therefore, it was not validated anymore for the other compounds.
Round 2
Reviewer 1 Report
After the re-submission of the manuscript answering referees' questions, it is suitable for publication in Molecules.